# Numerical and Analytical Studies of Soret-Driven Convection Flow Inside an Annular Horizontal Porous Cavity

**Abdelkader Mojtabi** [1,*] **, Khairi Sioud** [1,2] **, Alain Bergeon** [1] **and Marie Catherine Charrier-Mojtabi** [1]

[1] Institut de Mécanique des Fluides de Toulouse (IMFT), Université de Toulouse, CNRS,
31400 Toulouse, France; sioud.khayri@gmail.com (K.S.); Alain.bergeon@imft.fr (A.B.);
marie-catherine.mojtabi@univ-tlse3.fr (M.C.C.-M.)

[2] National Institute of Applied Science and Technology, Université de Carthage, La Marsa 2078, Tunisia

[*] Correspondence: mojtabi@imft.fr; Tel.: +33-5-61556793

**Abstract:** This paper studies the species separation of a binary fluid in a porous cavity between two horizontal concentric cylinders, submitted to a temperature gradient. The thickness of the cavity is $e = R_o - R_i$, where $R_i$ and $R_o$ are the internal and external radius, respectively. The numerous previous experiments performed in thermogravitational vertical columns (TGCs) showed that in order to obtain a significant separation, the thickness of the cell must be very small, compared with its height. Therefore, in our configuration, we considered $e \ll R_i$. The solution is assumed to be axisymmetric. Under the assumptions of parallel flow and forgotten effect, an analytical solution is obtained using Maple software, and the results are compared with those found numerically using Comsol Multiphysics. In natural convection, our results are in very good agreement with those evaluated with a regular perturbation method in powers of the dimensionless gap width $\varepsilon = \frac{e}{R_i}$ of order 15, and with the Galerkin method. The species separation calculated for our configuration is very close to the one obtained in a TGC column of height: $H = \pi R_i$. One of the main interests of the analytical solution presented here is that it can be used as a basic solution for a stability study analysis.

**Keywords:** Soret-driven convection; separation of binary fluid; porous medium; spectral method

## 1. Introduction

Natural convection in a horizontal annular cylinder filled with a monoconstituent fluid or with a porous medium saturated by a pure fluid has been widely studied [1–4]. Subsequently, the authors focused on the linear stability of the convective flow which arises regardless of the temperature difference between the inner and the outer cylinder. The energy stability of this flow was analyzed by Mojtabi and Caltagirone [5]. The approximate analytical solution of the convective flow, for low-temperature differences or low Rayleigh numbers, was obtained from a two- or three-order expansion of the Rayleigh number [1]. In 1987, Rao et al. [6] used a Galerkin method to obtain the convective flow solution. The approximation orders used by these authors did not exceed 20 in the radial and azimuthal directions. In 1988, Himasekhar and Bau [7] extended the regular perturbation method in powers of the filtration Rayleigh number up to $Ra^{60}$. In 1991, Charrier-Mojtabi et al. [8] numerically compared Fourier–Galerkin and collocation–Chebyshev methods and showed that the latter gave a more accurate description of the boundary layers near the inner and outer cylinders. In 1992, Mojtabi et al. [9] calculated the convective flow using a regular perturbation method in powers of the dimensionless gap $\varepsilon$.

$$\varepsilon = \frac{R_o - R_i}{R_i} = R - 1, \text{ where } R = \frac{R_o}{R_i}.$$

A symbolic algebra code allowed the authors to obtain the expanded solution at any order of $\varepsilon$ theoretically. In their article, the calculations were carried out up to order 7

because the computer used to run the symbolic software had limited available memory. The results calculated, at $O(7)$, for several radius ratios were compared with those obtained by the Galerkin method. From this comparison, it was found that the agreement between the results obtained by the two methods decreased when the ratio of the radii, R, or the Rayleigh number increased. We have recently extended the development to order $O(15)$. This will be the subject of the first part of this work.

Although the species separation has been extensively studied in vertical annular cylindrical columns [10] or inclined rectangular porous cells [11–16], the same cannot be said for the porous horizontal annular column, and, to our knowledge, the only reference is the paper of Abahri et al. [17], in which the authors obtained an approximate solution of the problem using a second-order expansion in Ra and a numerical solution using the Comsol software. In vertical thermogravitational columns, the extent of species separation is proportional to the height of the cavity. The distance between the two opposite isothermal, vertical or inclined surfaces leading to the optimum separation is less than 1 mm [11,12]. It follows that the annular cylindrical column should be of small thickness and large interior radius in order for the streamlines to be long enough to allow optimal species separation. For the study of species separation, it is, therefore, more relevant to write the solution of the problem in the form of an expansion in $\varepsilon = (R_o - R_i)/R_i \ll 1$ rather than an expansion in the Rayleigh, number, $R_a \gg 1$. The results in [9] show that the flow in the horizontal annular cylindrical column is almost conductive. In the continuation of this work, we seek to obtain a solution for the thermogravitational problem when the thermal field is purely conductive.

## 2. Mathematical Formulation

Thermodiffusion induces a mass fraction flux in binary fluid mixtures subjected to a temperature gradient. In addition to the usual expression for the mass flux density given by Fick's law, part of the expression due to the temperature gradient (namely, the Soret effect) is used so that [11,12]

$$\boldsymbol{J_m} = -\rho\, D^*\nabla C' - \rho\, C'(1 - C')D_T{}^*\nabla T', \tag{1}$$

where $\boldsymbol{J_m}$ is the vector density of mass flux. Darcy's law is assumed valid when binary fluid and solid phases are in local thermal equilibrium. It is also assumed that viscous dissipation, compressibility, and Dufour effects are negligible. The Boussinesq approximation is used as follows:

$$\rho = \rho_0[1 - \beta_T(T' - T_0) - \beta_C(C' - C_0)] \tag{2}$$

We also assume that the variation of $C'(1 - C')$ can be neglected, and this combination is replaced by $C_0(1 - C_0)$, where $C_0$ is the initial value of the mass fraction. Under these assumptions, the equations of continuity, momentum, energy, and species conservation are written in their dimensional form as follows:

$$\begin{cases} \nabla.\boldsymbol{V'} = 0 \\ \boldsymbol{V'} = -\frac{K}{\mu}\left(\nabla P' - \rho_0[1 - \beta_T(T' - T_0) - \beta_C(C' - C_0)]g\right) \\ (\rho c)^*\frac{\partial T'}{\partial t'} + (\rho c)_f\boldsymbol{V'}.\nabla T' = k^*\nabla^2 T' \\ \epsilon^*\frac{\partial C'}{\partial t'} + \boldsymbol{V'}.\nabla C' = D^*\nabla^2 C' + D_T{}^*C_0(1 - C_0)\nabla^2 T \end{cases}, \tag{3}$$

The boundary conditions for the horizontal annular cylindrical cell (Figure 1) are

$$\begin{cases} r = R_i, & T' = T_h, & \boldsymbol{V'}.\boldsymbol{n} = 0, & \boldsymbol{J_m}.\boldsymbol{n} = 0 \\ r = R_o, & T' = T_c, & \boldsymbol{V'}.\boldsymbol{n} = 0, & \boldsymbol{J_m}.\boldsymbol{n} = 0 \\ \theta = 0,\ \pi, & V'_\theta = \frac{\partial T'}{\partial \theta} = \frac{\partial C'}{\partial \theta} = 0 \end{cases} \tag{4}$$

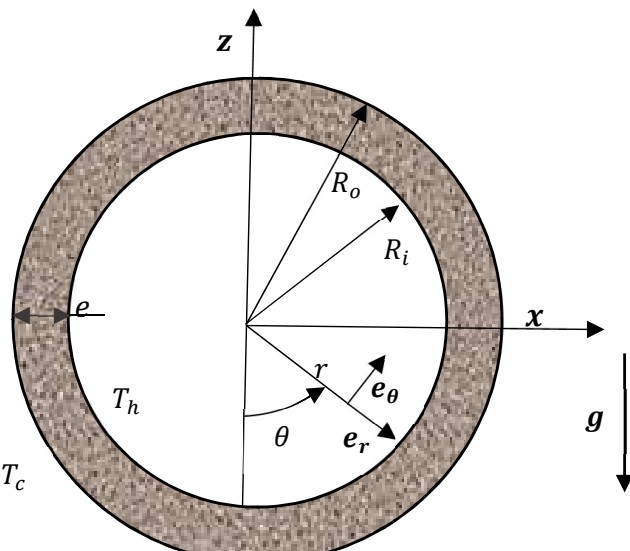

**Figure 1.** Schematic diagram of horizontal cylindrical TGC.

The system of equations is written in a dimensionless form by scaling length, time, velocity, temperature, and pressure, respectively, by: $R_i$, $\frac{(\rho c)^* R_i^2}{k^*}$, $\frac{k^*}{R_i(\rho c)_f}$, $(T_h - T_c)$, $\frac{k^* \mu}{K(\rho c)_f}$ and replacing $C' - C_0$ by $\Delta C = -\Delta T C_0 (1 - C_0) D_T^*/D^*$, where $D^*$ and $D_T^*$ are, respectively, the mass diffusion and thermodiffusion coefficient in a porous medium.

Finally, we obtain the following dimensionless system of equations:

$$
\begin{cases}
\nabla.\mathbf{V} = 0 \\
\mathbf{V} + \nabla P = Ra(\, T + \Psi\, C)\, \mathbf{z} \\
\frac{\partial T}{\partial t} + \mathbf{V}.\nabla T = \nabla^2 T \\
\epsilon \frac{\partial C}{\partial t} + \mathbf{V}.\nabla C = \frac{1}{Le}(\nabla^2 C - \nabla^2 T)
\end{cases}
, \qquad (5)
$$

and the associated dimensionless boundary conditions:

$$
\begin{cases}
r = 1, \;\; T = 1, \;\; \mathbf{V_r}.\mathbf{e_r} = 0, \;\; (\nabla C - \nabla T).\mathbf{e_r} = 0 \\
r = R, \;\; T = 0, \;\; \mathbf{V_r}.\mathbf{e_r} = 0, \;\; (\nabla C - \nabla T).\mathbf{e_r} = 0 \\
\theta = 0, \;\; \pi, \;\; V_\theta = \frac{\partial T}{\partial \theta} = \frac{\partial C}{\partial \theta} = 0
\end{cases}
, \qquad (6)
$$

where $\mathbf{z} = -\cos(\theta)\mathbf{e_r} + \sin(\theta)\mathbf{e_\theta}$.

The filtration Rayleigh number, Ra, and the separation factor, $\Psi$, are defined by $Ra = g\beta_T K R_i(\, T_h - T_c)(\rho c)_f/\nu k^*$ and $\Psi = -\beta_C D_T^* C_0(1 - C_0)/\beta_T D^*$. Here, $\nu = \mu/\rho$ is the kinematic viscosity of the mixture, $Le = \frac{k^*}{(\rho c)_f D^*}$ is the Lewis number, and $\epsilon = \epsilon^* \frac{(\rho c)_f}{(\rho c)^*}$ is the modified porosity.

## 3. Some Results in Natural Convection

In the present work, the development of the convective solution in the porous horizontal annular space was obtained at order 15 and extended the results previously obtained to order 7 in [9]. The thermal Rayleigh number, *Rath* is based on the thickness of the annulus, $(R_o - R_i)$, and not on the radius, $R_i$. *Rath* is deduced from *Ra* by the following relation: $Rath = Ra(R - 1) = \varepsilon Ra$. For this study, we made the following variable change $x = (r - 1)/\varepsilon$, so $x$ verifies $x \in [0, 1]$.

Table 1 presents the average Nusselt number, $Nu_g$, obtained from an expansion function of $\varepsilon$ at orders 3, 7, and 15 and compares it with those calculated by a Galerkin method for various values of *Rath* and for two values of the radius ratio ($2^{1/8}$ and $2^{1/4}$). For $2^{1/16}$,

the Galerkin method and approximations at orders 3, 7, 15 are in very good agreement for Rayleigh numbers ranging from 50 to $10^4$.

We observed that the global Nusselt number is a function of $Rath^2$, which is in agreement with the results of Himasekhar and Bau [7]. It is also interesting to note that the order 4 approximation obtained by Mojtabi [9]

$$Nu_g = 1 + 17Rath^2\left(\varepsilon^2 - \varepsilon^3\right)/40,320 + O\left(\varepsilon^4\right),$$

already provides a good approximation of the global Nusselt number. This approximation can be applied for a wide range of variations of the Rayleigh number. Table 1 shows that the approximation in $\varepsilon = \frac{R_o - R_i}{R_i} = R - 1$ at order 3 provides a very good approximation for low values of $\varepsilon$ and moderate values of Rayleigh number, $Rath$, (for $R = 2^{1/32}$, $2^{1/16}$, $2^{1/6}$, $2^{1/4}$, $\varepsilon \approx 0.0219$, 0.0443, 0.090, 0.189). The Nusselt number, $Nu_g$ remains close to 1.

**Table 1.** Average Nusselt number ($Nu_g - 1$) for different values of $Rath$ and R.

| *Rath* | Galerkin | *O*(15) | *O*(7) | *O*(3) |
|:---:|:---:|:---:|:---:|:---:|
| | | **R = 2$^{1/8}$** | | |
| $10^3$ | $252 \times 10^{-4}$ | $252 \times 10^{-4}$ | $252 \times 10^{-4}$ | $283 \times 10^{-4}$ |
| $2 \times 10^3$ | $946 \times 10^{-4}$ | $944 \times 10^{-4}$ | $951 \times 10^{-4}$ | $1132 \times 10^{-4}$ |
| $4 \times 10^3$ | $309 \times 10^{-3}$ | $352 \times 10^{-3}$ | $373 \times 10^{-3}$ | $453 \times 10^{-3}$ |
| | | **R = 2$^{1/4}$** | | |
| 200 | $177 \times 10^{-4}$ | $177 \times 10^{-4}$ | $177 \times 10^{-4}$ | $216 \times 10^{-4}$ |
| 500 | $102 \times 10^{-3}$ | $102 \times 10^{-3}$ | $102 \times 10^{-3}$ | $135 \times 10^{-3}$ |
| 750 | $208 \times 10^{-3}$ | $215 \times 10^{-3}$ | $232 \times 10^{-3}$ | $304 \times 10^{-3}$ |

## 4. Soret-Driven Convection: Species Separation

From Table 1, we deduce that the temperature field depends only on the radial coordinate $r$ as, for moderate Rayleigh numbers and for low values of $R$, the Nusselt number, $Nu_g$ remains close to 1, which corresponds to a pure conduction regime in the annular space. This result is obtained when the geometrical characteristics of the cavity correspond to the operating conditions, ensuring a good species separation: $e << R_i$. We assume that we are in the presence of a parallel convective flow. This means that apart from the neighborhood of $\theta = 0$, and $\theta = \pi$, the streamlines are almost arcs of circles parallel to the cavity boundaries. Under these conditions, we show that $|V_r| << |V_\theta|$ outside the neighborhood of $\theta = 0$, and $\theta = \pi$.

When $V_r << 1$ and $\frac{\partial T}{\partial \theta} = 0$, the energy equation reduces to

$$\nabla^2 T = 0$$

By taking the boundary conditions (6) into account, we obtain the following analytical expression of the temperature field:

$$T = 1 - \ln(r)/\ln(R) \tag{7}$$

By introducing the stream function, $\varphi$ defined by
$V_r = -\partial\varphi/r\partial\theta$ and $V_\theta = \partial\varphi/\partial r$, the continuity equation is automatically verified. The new formulation of the problem in dimensionless form, with the hypothesis of the forgotten effect, previously used by Furry Jones and Onsager (FJO, theory) [18] is given by

$$\nabla^2\varphi = Ra\ \sin(\theta)\frac{\partial T}{\partial r} \tag{8}$$

The conservation equation of mass fraction is given by

$$\frac{\nabla^2 C}{Le} = V_r \frac{\partial C}{\partial r} + V_\theta \frac{\partial C}{r\partial\theta} \tag{9}$$

Setting $\varphi = Ra \sin(\theta) f(r)$ Equation (8) leads to

$$\frac{\partial\left(r\frac{\partial f}{\partial r}\right)}{\partial r} - \frac{f}{r} - r\frac{\partial T}{\partial r} = 0 \tag{10}$$

The circles of radius $R_i$ and $R_o$ connected by the vertical segments $\theta = 0$ and $\theta = \pi$ forms a streamline associated with $\varphi = 0$, and it implies that $f(r = 1) = 0$ and $f(r = R) = 0$.

By taking these two boundary conditions into account in the differential Equation (10), we have the following analytical expression of the solution

$$f(r) = \frac{R^2\left(r^2 - 1\right)\ln(R) - r^2\left(R^2 - 1\right)\ln(r)}{2r(R^2 - 1)\ln(R)} \tag{11}$$

From the expression of $f(r)$, we deduce the analytical expression of the two components of the velocity

$$\begin{cases} V_{ra} = -Ra\, f(r)\cos(\theta)/r \\ V_{\theta a} = \frac{M}{2r^2(R^2-1)\ln(R)} \end{cases}, \tag{12}$$

where the numerator M is given by

$$M = Ra\,\sin(\theta)\left(R^2\left(r^2 + 1\right)\ln(R) - r^2\left(R^2 - 1\right)(\ln(r) + 1)\right) \tag{13}$$

Figure 2, shows that the variations of the stream function $\varphi(r,\theta)$ with $r$ for $\theta = \pi/2$, $Ra = 100$ and $R = 2^{1/16}$ obtained analytically and from direct numerical simulations, using Comsol Multiphysics software, are in very good agreement. The solid brown line is taken from the analytical results, and the dots in black represent the results of the direct numerical simulation.

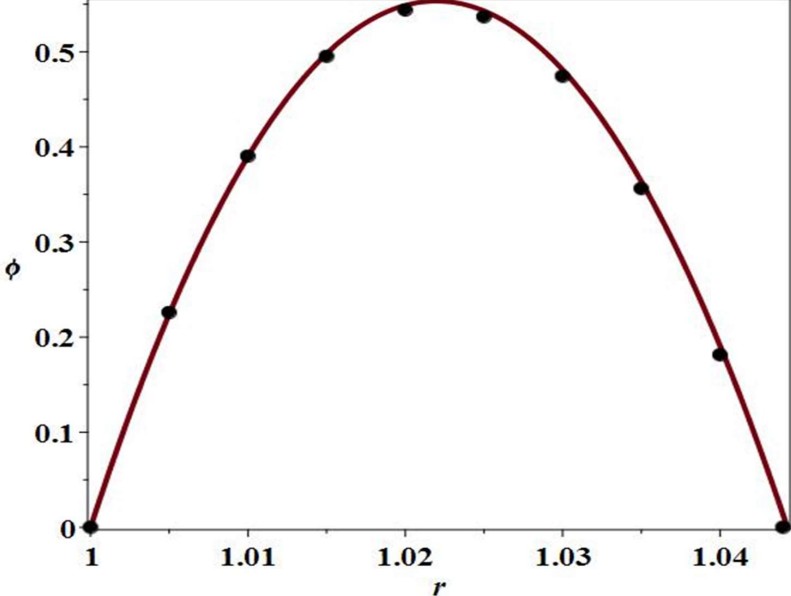

**Figure 2.** Stream function $\varphi(r, \theta = \pi/2)$ for $Ra = 100$ and $R = 2^{1/16}$ obtained analytically (continuous line) and using numerical direct simulation (black dots).

The analytical results obtained for low values of the radius ratio, $R$, and moderate values of the thermal Rayleigh number, $Ra$, show that $V_r \ll V_\theta$.

In Figure 3, $V_\theta(r, \theta = \pi/2)$ is presented as a function of $r$ for $Ra = 100$ and $R = 2^{1/16}$.

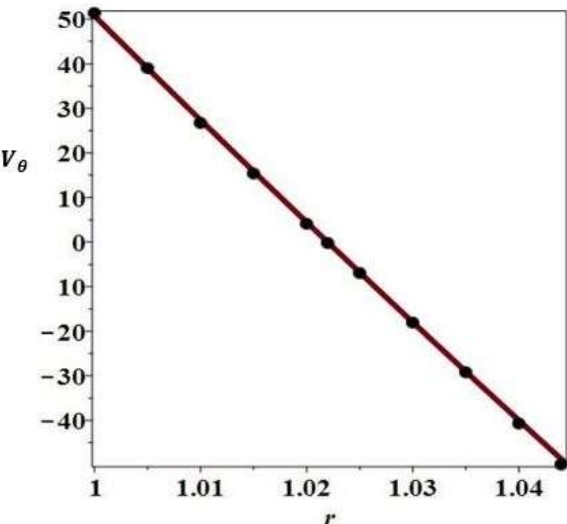

**Figure 3.** $V_\theta$ versus $r$ for $Ra = 100$ and $R = 2^{1/16}$ obtained analytically (continuous line) and using numerical direct simulations (black dots).

In Figure 4, $V_\theta(r, \theta = \frac{\pi}{2})$ and the norm of the gradient of the stream function, $|\nabla \varphi|$ are presented as a function of $r$ for $Ra = 100$ and $R = 2^{1/16}$. From this figure, we deduce that $V_r \ll V_\theta$, since: $|\nabla \varphi| \cong V_\theta (1 + (\frac{V_r}{V_\theta})^2)^{1/2} V_\theta (r, \theta = \frac{\pi}{2})$ is drawn in brown and $|\nabla \varphi|$, in black. Figure 4 shows indirectly that $|\nabla \varphi| \cong V_\theta$. The dots in black are obtained from the direct numerical simulation.

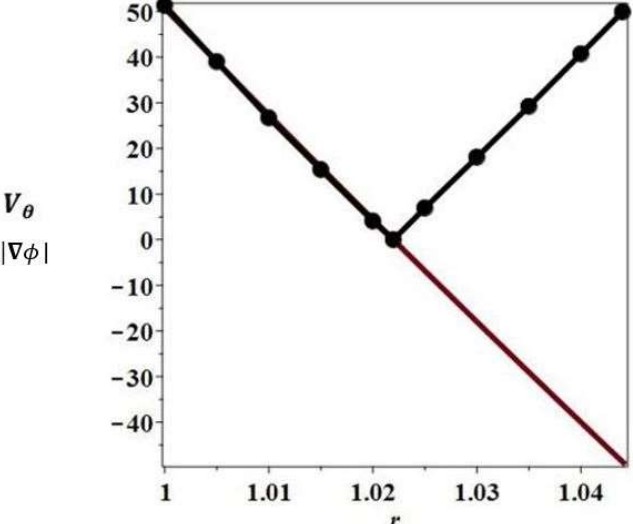

**Figure 4.** $V_\theta$ and $|\nabla \varphi|$ versus $r$ for $\theta = \frac{\pi}{2}$, $Ra = 100$ and $R = 2^{1/16}$ obtained analytically (continuous line) and using numerical simulations (black dots).

The conservation equation of mass fraction (9) reduces to a linear equation of the second order as follows:

$$\frac{\partial \left(r \frac{\partial C}{\partial r}\right)}{\partial r} + \frac{\partial^2 C}{r \partial^2 \theta} = Le\left(r V_r \frac{\partial C}{\partial r} + V_\theta \frac{\partial C}{\partial \theta}\right), \tag{14}$$

with associated boundary conditions:

$$
\begin{cases}
r = 1, & \dfrac{\partial C}{\partial r} + \dfrac{1}{r\ln(R)} = 0 \\
r = R, & \dfrac{\partial C}{\partial r} + \dfrac{1}{r\ln(R)} = 0 \\
\theta = 0, \quad \pi, & \dfrac{\partial C}{\partial \theta} = 0
\end{cases}
\tag{15}
$$

Although $V_r \ll V_\theta$ in the central part of the horizontal annular column, this inequality is not valid for $\theta$ in the vicinity of 0 and $\pi$ since $V_\theta = 0$ for $\theta = 0$ and for $\theta = \pi$.

Therefore, it was not possible to obtain an explicit analytical relation for the mass fraction $C$ solution of Equation (14). On the other hand, we replaced $V_r$ and $V_\theta$ by their expressions $V_{ra}$ and $V_{\theta a}$ given by the system of Equation (12) to obtain an equation where the only unknown is $C$, which we solved numerically. Then, comparing the values of the mass fraction $C$ obtained from direct numerical simulation (resolution of the system with four conservation Equation (5), without simplifying assumption) and of the numerical resolution of Equation (16) alone, with only $C$ as unknown, associated with boundary conditions (15), we have

$$
\frac{\partial\left(r\frac{\partial C}{\partial r}\right)}{\partial r} + \frac{\partial^2 C}{r\partial^2\theta} = Le\left(rV_{ra}\frac{\partial C}{\partial r} + V_{\theta a}\frac{\partial C}{\partial \theta}\right)
\tag{16}
$$

The results of direct numerical simulation and of the solution obtained from Equation (16) alone are compared in Figure 5.

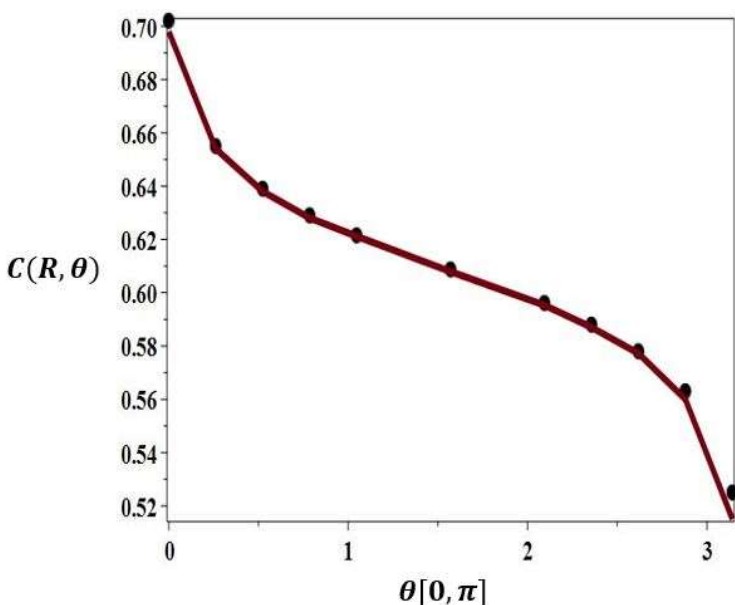

**Figure 5.** $C$ ($r = R$, $\theta$) versus $\theta$ for $Ra = 12.6$, $Le = 100$, $\Psi = 0.21$ and $R = 2^{1/16}$ obtained analytically (continuous line) and using numerical direct simulation (black dots).

The evolution of $C$ as a function of $\theta$ for $r = R$ shows that the two approaches are perfectly concordant and that, in the central part of the horizontal column, the evolution of $C$ as a function of $\theta$ is linear for any value of $r \in [1, R]$.

Figure 6 illustrates the variation in the mass fraction field obtained numerically for $Ra = 50$, $Le = 100$, $\Psi = 0.2$ and $R = 2^{1/4}$. The colored scale represents the intensity of the mass fraction of the heaviest component. The lines represent the associated streamlines.

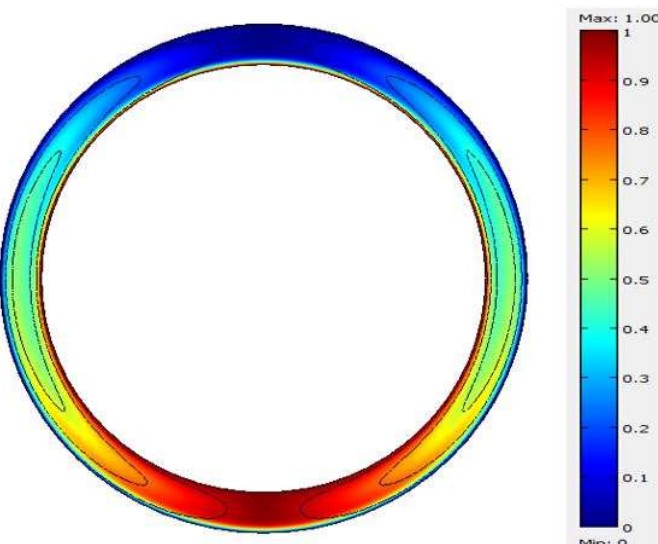

**Figure 6.** Variation in the mass fraction for $Ra = 50$, $Le = 100$, $\Psi = 0.2$ and $R = 2^{1/4}$ blue (red) corresponds to low (high) mass fraction values.

Under the effect of thermodiffusion, the convective regime within the annular column leads to a low value of $C$ (blue) at the top of the column and a higher value of $C$ (red) at its lower part. The difference between these two extreme values of $C$ indicates the degree of species separation of the mixture.

## 5. Conclusions

Recent developments in the field of symbolic computation have made it possible to compute very accurate analytical solutions to non-linear problems. The analytical solution obtained previously to order 7, using a regular asymptotic development in powers of in $\varepsilon = \frac{R_o - R_i}{R_i} \ll 1$, was extended here to order 15. The solution obtained for the problem of natural convection in a horizontal porous annular cylinder for a wide range of Rayleigh numbers could be considered as a reference solution for the validation of future numerical 2D codes.

However, for the study of the species separation in a binary fluid, this solution sought in the development of $\varepsilon$ is not convenient for carrying out a stability analysis or obtaining analytically, in this case, the optimum of species separation. In the case of the parallel flow approximation and forgotten effect, we showed good agreement between the direct numerical simulation and the analytical expression of the convective flow. We established that the radial velocity component is negligible, compared with the tangential component, except in the vicinity of $\theta = 0, \pi$. From the velocity and temperature fields, we were able to calculate the mass fraction, $C(r, \theta)$, using a numerical resolution of a single scalar equation with partial derivatives. This solution is in good agreement with the results of direct numerical simulations.

We obtained for the first time an analytical solution of Soret-driven convection flow in an annular, thin, horizontal porous layer saturated with a binary fluid.

Until now, the species separation was carried out in parallelipipedic or annular vertical columns. In this study, we showed that the species separation of a binary mixture can also be obtained in a horizontal porous annular column.

**Author Contributions:** Conceptualization, A.M., methodology, software, validation, writing, A.M., K.S., A.B., M.C.C.-M., All authors have read and agreed to the published version of the manuscript.

**Funding:** This research received no external funding.

**Institutional Review Board Statement:** Not applicable.

**Informed Consent Statement:** Not applicable.

**Data Availability Statement:** Exclude this statement.

**Acknowledgments:** This work was supported by CNES, the French National Space Agency.

**Conflicts of Interest:** The authors declare no conflict of interest.

## Nomenclature

| | |
|---|---|
| $C$ | Mass fraction |
| $C_0$ | Initial mass fraction |
| $D$ | Mass diffusion coefficient, $[\text{m}^2\text{s}^{-1}]$ |
| $D^*$ | Mass diffusion coefficient in a porous medium $[\text{m}^2\text{s}^{-1}]$ |
| $D_T^*$ | Thermodiffusion coefficient in porous medium $[\text{m}^2\text{s}^{-1}\text{K}^{-1}]$. |
| $e$ | Thickness of the two cells [m] |
| $g$ | Gravitational acceleration $[\text{ms}^{-2}]$ |
| $H$ | Height of the cavity [m] |
| $K$ | Permeability of porous medium $[\text{m}^2]$ |
| $k^*$ | Effective thermal conductivity [W/m K] |
| $Le$ | Lewis number |
| $P'$ | Pressure [Pa] |
| $R$ | Radius ratio, $R = \frac{R_o}{R_i}$ |
| $Ra$ | Filtration Rayleigh number, $Ra = g\beta_T K R_i \left( T_h - T_c \right)(\rho c)_f / \nu k^*$ |
| $Rath$ | $Rath = Ra\,(R-1) = \varepsilon Ra$ |
| $R_i$ | Inner sphere radius [m] |
| $R_o$ | Outer sphere radius [m] |
| $t'$ | Time, [s] |
| $T'$ | Temperature [K] |
| $T_c$ | Cold temperature [K] |
| $T_h$ | Hot temperature [K] |
| $T_0$ | Initial temperature [K] |
| $V_r$ | Radial velocity |
| $V_\theta$ | Tangential velocity |
| *Greek symbols* | |
| $\beta_T$ | Thermal expansion coefficient, $[\text{K}^{-1}]$ |
| $\beta_C$ | Solutal expansion coefficient |
| $\varepsilon$ | Dimensionless gap, $\varepsilon = R - 1$ |
| $\varepsilon^*$ | Porosity of the porous medium |
| $\epsilon$ | Modified porosity |
| $\mu$ | Dynamic viscosity of the mixture, $[\text{m}^2\text{s}^{-1}]$ |
| $\rho$ | Density of the binary fluid $[\text{kg/m}^3]$ |

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
