# Peer review of "Numerical and Analytical Studies of Soret-Driven Convection Flow Inside an Annular Horizontal Porous Cavity"

_fluids, doi:10.3390/fluids6100357_

Round 1

Reviewer 1 Report

Manuscript number: fluids-1365371-v1

Title:  Numerical and analytical studies of Soret-driven convection flow inside an annular horizontal porous cavity

The numerical and analytical study of the convective flow that arises within a porous medium between two coaxial cylinders, which surfaces are maintained at different temperatures. A system of equations governs the fluid flow, heat transfer and the mass fraction transport. The authors use a perturbation method with series in powers of the gap size in order to obtain a solution of the problem; its development involves symbolic calculations. The predictions of the Nusselt number are compared with results calculated by a Galerkin method. An approximate analytical solution is obtained for the case of a very narrow gap.

This is an interesting study that contributes in the progress of this domain of the research in porous media. The exposition of the material, the formulation of the problem and the presentation of methods and results are satisfactory, the conclusions of the article are justified by the findings. I have only some minor remarks.

1) P.4, above eq.(1), what does it signify, 'the usual isothermal contribution '? The temperature is not the same in various places of sample, so the expression is not clear.

2) P.4, above eq.(1), is it correct that 'Thermo-diffusion induces a mass fraction gradient...'? May be it is appropriate to state that it induces a mass fraction flux, as it can be seen from eq.(1)?

3) P.4, below eq.(2), is the statement 'the variation of ?′(1−?′) can be replaced by ?0(1−?0)...' exact? Do the authors mean, 'the variation of ?′(1−?′) can be neglected and this combination is replaced by ?0(1−?0)...'?

4) P.7, 1st line, 'From Table 1, we deduce that the temperature field depends only on the radial coordinate r'; this is not clear how this conclusion is obtained. Some comments on this point are necessary.

5) The legend to Fig.4 is not quite clear: how are distinguished the velocity Vtet and |grad phi|? Probably by line colors? The dotted line (in fact, black symbols connected by a line) presents only |grad phi| and not Vtet. The authors should clarify this point.

The manuscript deserves to be published in Fluids after some minor revision.

Reviewer 2 Report

The topic fits into the journal, and some works have been done. The paper is only at the boundary of pass. 

[1] The writing is at very low level. See how many writing issues. For example, many sentences and figures twisted positions. Figures lack caption, legend, etc. Variables should be defined  when they first appear, in italic format but the numeric values and units should be in normal font. Consider using MDPI editing service.

[2] Each equation should be numbered separately, but not together. 

[3] Model validations are too limited. Try to do some error analysis to quantify the errors to see how far it is between your results and benchmarks.  

[4] Porous concept is in your title, so you should heavily discuss its effect, such as, if you change porosity or permeability, etc how would your results change. In present paper, you mostly mentioned your results are better but lack critical discussions on underlying physical mechanics on the problem you investigated. 

[4] I will not require you to do extra calculation cases, but you need to enrich your current case study in good way.

Reviewer 3 Report

Comments to the Author

Manuscript fluids-1365371:

Numerical and analytical studies of Soret-driven convection flow inside an annular horizontal porous cavity

by Abdelkader Mojtabi , Khairy Sioud , Alain Bergeon , Marie-Catherine Mojtabi

The authors present a comparison between an analytical solution and a numerical model for the problem of natural convection in horizontal porous cylindrical annular. The main result is the improvement of the analytical solution, extended to order 15 whit respect to their previous work dated 1992. 

The work is well written, consistent and of some use for the scientific community. Therefore, I have just some minor remarks that maybe can they can make the work more engaging and accessible for readers less familiar with the subject.

  •  I suggest in the Introduction to write a paragraph explaining where, applicatively speaking, the study of flows in such cavities is useful.
  • page 1, fourth line from the bottom: the reference is wrong, it should be written 'Mojtabi & Caltagirone'
  • page 6 ninth line from the top: a symbol is missing
  • Figure 2, 3, 4, 5: in the caption is written 'dottle line) but, except for figure 4, I see only dots. Furthermore, I think that dots are more appropriate, otherwise, as in figure 4, it is difficult to see the two lines. In figure 4, it is not clear to me the deviation between the black and red lines, why this occurs? Maybe the authors can explain it in the text. 
  • Figure 6, a colour bar is missing.
  • It is not clear to me what the Autors mean with binary fluid: is it a multiphase flow? Are there two fluids? which are the fluids involved? 
  • It is not clear to me what the Autors mean with mass fraction file, explain in the text. 

Round 2

Reviewer 2 Report

Ok, I agree the acceptance of the paper. As an experienced referee who has reviewed more than 1000 papers, next time I would not approve your response letter easily.

By the way, the authors should note their writing styles. For example, “The energy stability of this flow was analyzed by Mojtabi and Caltagirone [5]” takes one paragraph. These can be fine-tuned in proof stage easily.
